# B7-H3 in Medulloblastoma-Derived Exosomes; A Novel Tumorigenic Role

**DOI:** 10.3390/ijms21197050

**Published:** 2020-09-25

**Authors:** Ian J. Purvis, Kiran K. Velpula, Maheedhara R. Guda, Daniel Nguyen, Andrew J. Tsung, Swapna Asuthkar

**Affiliations:** 1Departments of Cancer Biology and Pharmacology, University of Illinois College of Medicine Peoria, Peoria, IL 61605, USA; Ian.Purvis@cuanschutz.edu (I.J.P.); velpula@uic.edu (K.K.V.); gmreddy@uic.edu (M.R.G.); dnguye68@uic.edu (D.N.); Andrew.J.Tsung@ini.org (A.J.T.); 2Departments of Neurosurgery, University of Illinois College of Medicine Peoria, Peoria, IL 61605, USA; 3Departments of Pediatrics, University of Illinois College of Medicine Peoria, Peoria, IL 61605, USA

**Keywords:** medulloblastoma (MB), extracellular vesicle (EV), exosomes, B7-H3

## Abstract

(1) Aim: Medulloblastoma is the most common aggressive pediatric cancer of the central nervous system. Improved therapies are necessary to improve life outcomes for medulloblastoma patients. Exosomes are a subset of extracellular vesicles that are excreted outside of the cell, and can transport nucleic acids and proteins from donor cells to nearby recipient cells of the same or dissimilar tissues. Few publications exist exploring the role that exosomes play in medulloblastoma pathogenesis. In this study, we found B7-H3, an immunosuppressive immune checkpoint, present in D283 cell-derived exosomes. (2) Methods: Utilizing mass spectrometry and immunoblotting, the presence of B7-H3 in D283 control and B7-H3 overexpressing exosomes was confirmed. Exosomes were isolated by Systems Biosciences from cultured cells as well as with an isolation kit that included ultracentrifugation steps. Overlay experiments were performed to determine mechanistic impact of exosomes on recipient cells by incubating isolated exosomes in serum-free media with target cells. Impact of D283 exosome incubation on endothelial and UW228 medulloblastoma cells was assessed by immunoblotting. Immunocytochemistry was employed to visualize exosome fusion with recipient cells. (3) Results: Overexpressing B7-H3 in D283 cells increases exosomal production and size distribution. Mass spectrometry revealed a host of novel, pathogenic molecules associated with B7-H3 in these exosomes including STAT3, CCL5, MMP9, and PI3K pathway molecules. Additionally, endothelial and UW228 cells incubated with D283-derived B7-H3-overexpressing exosomes induced B7-H3 expression while pSTAT1 levels decreased in UW228 cells. (4) Conclusions: In total, our results reveal a novel role in exosome production and packaging for B7-H3 that may contribute to medulloblastoma progression.

## 1. Introduction

In recent years, the role of extracellular vesicles (EVs) in tumor progression has become an interesting area of research for understanding and combating cancer. In cancer, EVs mediate the transfer of oncogenic proteins, nucleic acids, and lipids from tumor cells to other tumor cells and the surrounding microenvironment [1,2,3]. Recent classification of EVs has led to distinctions based on size (exosomes: 30–100 nm, microvesicles: 100–1000 nm, oncosomes: 1000–10,000 nm) [4]. The differing sizes indicate different methods for EV production such that larger EVs form from blebbing of the cellular plasma membrane whereas smaller exosomes likely originate from multivesicular bodies within cells, carrying with them nucleic acids, tetraspanins, and other proteins including fibronectin, P-gp, LRG1, HER2, and c-Met [5,6,7]. In cancer, this process is believed to impart pro-tumorigenic properties to the local tumor microenvironment, facilitating functions such as immune evasion, angiogenesis, metastasis, and drug-resistance to neighboring cells.

Breast cancer cells have been shown secreting exosomes loaded with PD-L1. These exosomes could induce PD-L1 expression in cell lines low in PD-L1, leading to immune evasive activity [8]. Exosomes isolated from pancreatic cancer patient serum were enriched in glypican-1, a useful biomarker for predicting pancreatic cancer [9]. Interestingly, the tumor microenvironment can influence nontumor cells to help maintain a pro-tumorigenic milieu by changing the composition of their EVs. For instance, tumor-associated macrophages (TAMs) were shown secreting EVs packaged with a long noncoding RNA (HISLA) that stabilized HIF-1α in breast cancer cells. This helped the tumor cells maintain glycolysis while also promoting chemoresistance [10]. The authors showed that the composition of the TAM-derived EVs was likely maintained by signaling mechanisms from the breast cancer cells through lactate production, leading to increased ERK-ELK1 signaling.

No exception to these examples, brain tumors also utilize EVs to maintain their environment and spread to distal tissue regions. EGFRvIII was found in EVs isolated from glioblastoma cells, and these EVs were able to upregulate the expression of EGFRvIII target genes, such as VEGF, in another glioma cell line [11]. EVs containing Cre recombinase have been developed and used in a murine glioma model to trace EV mRNA dissemination in vivo, implicating EVs as mediators of cancer pathogenicity [12].

To date, little is known about the role of exosomes in medulloblastoma (MB) progression. Studies have indicated pro-tumorigenic roles of MB-derived exosomes including promotion of proliferation and metastasis in MB cells, but no specific mechanisms of action or associated signaling cascades have been fully elucidated [13,14,15]. Further characterization of exosomes in MB may lead to better understanding of subgroup-specific heterogeneity and help explain the differing propensity to disseminate locally and distally between MB subtypes. With the promise of liquid biopsies and similar detection techniques in early tumor recognition, further research into the nature of MB-derived exosomes may even lead to the discovery of new MB biomarkers that could lead to the production of quicker and less expensive detection methods [16].

In our previous study, we showed a novel role for B7-H3, an immune suppressive ligand whose receptor is unknown, in MB angiogenesis, but the full role of B7-H3 in MB progression is still unclear [17]. To further characterize the role of B7-H3 in MB progression, we conducted various analyses on B7-H3-containing exosomes derived from MB cell lines. This study shows that B7-H3 is packaged in MB-derived exosomes [18]. Overexpression of B7-H3 in MB cells not only increased the presence of B7-H3 protein in exosomes but also the size and concentration of exosomes. Additionally, mass spectrometry revealed various proteins packaged along with B7-H3 that may suggest novel tumorigenic pathways influenced by B7-H3. These results suggest a novel pathogenic role for B7-H3 that may be independent of any potential B7-H3 receptors. Our results imply a new function for B7-H3 in MB progression while providing further evidence for the importance of investigating exosome-related pathways in MB. Collectively, these results imply that B7-H3 present in cancer cell exosomes may play an important role in cell–cell communication and tumor cell signaling in addition to its role in immuno-evasion.

## 2. Results

### 2.1. Analysis of B7-H3 in MB Exosomes

PD-L1 has been found in cancer cell exosomes and can promote immune evasion [8]. PD-L1 has also been found in exosomes derived from various cancers, exhibiting protumorigenic functions without the need for interaction with its receptor, PD-1 [8,19,20]. PD-L1 and B7-H3 belong to the B7-CD28 superfamily of genes, and both cell surface ligand proteins have immunosuppressive functions [21,22]. Recent studies have shown that B7-H3 also has a cleaved, soluble form that may play a role in tumor angiogenesis [23,24,25]. Despite these known roles, no publications to date have investigated the possible influence that B7-H3 may have on exosomal production or packaging in MB. To evaluate the presence of B7-H3 in the exosomes, we first verified the baseline cellular levels of B7-H3 in the control and B7-H3 overexpressing (B7-H3 OE) D283 cells using immunoblot. We observed a significant (*p* < 0.001) increase in the levels of B7-H3 in B7-H3 OE cells compared to the control (Appendix A). Exosomes were isolated from the conditioned media of both control and B7-H3 OE D283 cells and the purity of the exosomal fractions were analyzed by fluorescent NTA (Figure 1A). The pure exosomal fraction confirmed by System Biosciences (SBI) was used in further analyses. The conditioned media from B7-H3 OE showed nearly a 2-fold increase in the absolute concentration of exosomes when compared to exosomes obtained from control cells (Figure 1A,B). The fluorescence peak in Figure 1A indicates that the purified fraction is enriched by particles below 100 nm in diameter, indicating a purified exosome fraction. This purified exosome fraction was then subjected to mass spectrometry and coculture experiments in this study. Mass spectrometry analysis of exosomes showed the presence of B7-H3 and other known exosome markers such as CD63, CD9, and TSG101 (Figure 1C and Table 1). Interestingly, B7-H3 has also been found in neuroblastoma-derived exosomes, another solid pediatric tumor [26]. Our data indicate that B7-H3 overexpressing MB cells have enhanced ability to secrete B7-H3 into exosomes, indicating a potential role for B7-H3 in MB exosome production.

### 2.2. B7-H3 Overexpressing Exosomes Contain Novel Pro-Tumorigenic Molecules

To analyze the various molecules associated with B7-H3, exosomes from the control and B7-H3 OE D283 cells were analyzed by LC-MS/MS. The protein hits from the LC-MS/MS were then uploaded to PANTHER (pantherdb.org), a web based comprehensive tool that annotated the protein hits into associated signaling pathways and protein class based on functions (Figure 2). Unsurprisingly, molecules associated with chemokine and cytokine signaling pathways represented the largest fraction of detected proteins (18%) in B7-H3 OE exosomes. Interleukin signaling molecules also comprised a notable portion of hits from this analysis (10%). Interestingly, exosomes included JAK/STAT (7%), angiogenesis-associated molecules (7%), PDGF (13%), PI3K (7%), and glycolysis-associated (5%) signaling molecules (Figure 2A).

We also observed that defense/immune-related proteins (1%) were present in a much lower percentage than other classes of proteins such as hydrolases (10%), nucleic acid binding proteins (16%), transcription factors (7%), and enzyme modulators (6%). Membrane proteins constituted an expectedly large percentage of detected proteins (9%) (Figure 2B).

Additionally, Table 1 highlights shared and distinct molecules found in exosomes derived from control or B7-H3 OE D283 cells. We found that both exosome groups have exosome markers such as CD9 but TSG101 was only found in B7-H3 OE exosomes. CD9 is a tetraspanin, widely considered an exosome marker in various cell types including cancer cells [5,27,28,29]. TSG101 is an important component of the ESCRT-0 complex that promotes exosome secretion. Due to its essential role in exosome secretion, TSG101 may be considered an exosome marker that helps distinguish exosomes from other EVs [30]. Both groups of exosomes also contained HS90A, consistent with previous observations regarding MB exosome content [13]. Interestingly, recent studies have shown HSP90 may play a role in exosome biogenesis and secretion, and that other heat-shock proteins may also be utilized by cancer cells to avoid apoptosis and promote angiogenesis [31,32,33]. Notably, we observed a greater number of MMP isoforms in the B7-H3 OE exosomes, including MMP-2 and MMP-9, than control exosomes. A relationship between B7-H3 and these major MMPs has been observed previously, implicating B7-H3 in the angiogenesis and metastasis processes of tumor progression [34,35]. To our knowledge, this is the first report linking B7-H3 directly with MMPs via their presence in MB-derived exosomes. Interestingly, control exosomes contained STAT1 while B7-H3 OE exosomes contained STAT3. B7-H3 has recently been associated with increased STAT3 activity, leading to further angiogenesis in tumor cells [36]. These results suggest that B7-H3 may play an active role in exosome biogenesis and exosome-mediated pathogenicity in MB.

### 2.3. B7-H3 Overexpressing Exosomes Translocate into Stromal and Cancer Cells

Our results indicate that B7-H3 may play an active role in exosome biogenesis and exosome-mediated pathogenicity in MB. To investigate a more mechanistic role of B7-H3 in exosomes, we performed overlaying experiments. Immunoblotting of the isolated exosomes revealed a 110% increase of B7-H3 levels in B7-H3 OE cells when compared to control D283 cells (*p* < 0.01) (Figure 3A). Additionally, calcein AM green fluorescence was used to stain D283 B7-H3 OE exosomes in bulk and were incubated with D283, D425, and D458 MB cells. F-actin (red) staining was used to visualize the actin filaments of MB cells. While actin is an intracellular filament, F-actin tends to localize directly beneath the surface of the cell membrane, lending itself as a marker for cell membrane colocalization studies [37,38,39]. We chose F-actin as it is used for staining in adherent cells due to its association with cell membranes and adherent proteins [40,41]. The exosomes were found localizing near F-actin stained microfilament networks of MB cells, indicating the possible association of B7-H3 OE exosomes with MB cancer cells (Figure 3B). The negative control for nonspecific Calcein AM staining is represented in the Appendix A. Previously we reported on the ability of soluble B7-H3 produced by D283 cells in promoting angiogenesis in endothelial cells [17]. To further validate if these properties of D283 exosomes could influence endothelial cells, and whether B7-H3 plays a role in this process, HMECs were incubated with exosomes derived from D283. Western blotting showed that B7-H3 OE exosomes were able to increase endogenous B7-H3 levels in HMECs by 467% (*p* < 0.001) (Figure 3C) suggesting B7-H3 may influence the tumor microenvironment through exosomes. UW228 cells, with lower Myc [42] and B7-H3 levels (Appendix A) when incubated with D283 exosomes showed increased B7-H3 while showing a 48% decrease (*p* < 0.001) in p-STAT1 levels (Figure 3D). These results suggest a receptor-independent role of B7-H3 that may indicate distal and proximal roles of B7-H3 in the cancer microenvironment.

## 3. Discussion

Research into the role of B7-H3 in cancer pathogenesis has primarily focused on its function as a receptor with some evidence highlighting its role as a soluble, nonmembrane bound ligand [18,23]. This study has furthered our understanding of plausible pathogenic functions of B7-H3 and has identified a new avenue by which B7-H3 may enact its tumor-promoting functions through exosome packaging and production. These findings are particularly significant for MB research as little is known about MB-derived exosomal packaging, production, or their contents. Exosomes are known to contribute to tumor progression, and possible initiation, by shuttling oncogenes and oncoproteins from tumor cells to local and distal cells [6,43,44]. The contents of cancer cell exosomes, and the molecules associated with B7-H3 in MB-derived exosomes, could provide valuable insight into how B7-H3-packaged exosomes contribute to MB progression. For instance, previous studies have indicated a link between B7-H3 expression and MMP2/9 activity and expression [18,45]. This association may be one such mechanism by which B7-H3 promotes tumor migration, angiogenesis, and overall aggression. Another study observed a relationship between B7-H3 and the PI3K/AKT signaling pathway. The authors showed that B7-H3 may activate the PI3K/AKT pathway, leading to STAT3 phosphorylation and subsequent MMP transcription [36]. As observed in our mass spectrometry data and previous data, B7-H3 may have a relationship with chemokines such as CCL5 and CCR5 [17]. Interaction between CCL5 and CCR5 can induce PI3K/AKT activity [46]. CCL5/CCR5 interactions have also been shown to promote cancer cell migration, possibly by upregulating MMP production [47]. Observations from our data suggest a strong chemokine relationship with B7-H3 that may provide a mechanistic explanation for the correlation between B7-H3 and MMP2/9 expression seen in previous reports. Future studies may aim to characterize specifically how the intracellular domain of B7-H3 interacts with mediators of PI3K/AKT activation and downstream targets of this signaling pathway.

A relationship between B7-H3 and PI3K/AKT, while hinted at in various cancer studies, has not been fully established [36,48,49,50]. Most of these studies suggest that B7-H3 may act upstream of PI3K/AKT activation in tumor cells, leading to the activation of genes associated with chemoresistance, metastasis, cell proliferation, and glycolysis. Our current study adds another dimension that may help further our understanding of this potential signaling relationship; PI3K/AKT activation may help mediate and/or promote the observed roles of B7-H3 in exosome formation and secretion. Our analysis revealed that PI3K/AKT pathway associated proteins comprise a noteworthy portion of proteins found in B7-H3 OE exosomes (7%) including isoforms of PI3KC2. PIK3C2 are class II PI3K enzymes that are associated with the plasma membrane and Golgi apparatus [51,52]. Studies have shown that PI3KC2 enzymes play various roles in facilitating cell migration, receptor signaling transduction and endocytosis, as well as potential functions in Golgi-associated vesicle formation and secretion [51,52]. Additionally, previous studies have indicated that PI3KC2 isoforms may be necessary components for proper internalization and signal transduction of major tumorigenic receptors such as VEGFR2 and TGF-β [53,54]. Future investigations could focus on determining if Class II PI3Ks are necessary for proper B7-H3 tumorigenic signaling. Interestingly, Class II PI3Ks have been implicated in proper vesicle formation and secretion although more work is needed to verify previous observations [55]. Lipid products from Class II PI3K enzymatic activity are used to form intracellular vesicles, some of which may be excreted by cells [56]. Our mass spectrometry data revealed the presence of PI3K-C2β and PI3K-C2γ isoforms in B7-H3 OE exosomes. Based on their potential roles in vesicle formation and trafficking, by integrating these proteins into MB exosomes, these molecules may help establish a feed forward mechanism, increasing the likelihood of inducing elevated oncogenic exosome production in recipient cells, possibly in a B7-H3-dependent manner. If and how B7-H3 may influence the expression or activity of these mediators in relevant tumorigenic contexts remains to be established.

Our previous study showed that STAT1 may have a tumor suppressive role in MB [17]. As described above, STAT3 may have tumor promoting activity in MB by upregulating prometastatic molecules. These observations are echoed by previous literature suggesting that generally STAT1 may be considered a tumor suppressor while STAT3 acts as a tumor promoter, although more research is needed to outline the specific instances where this characterization holds true [57]. Prior studies have shown that JAK/STAT signaling is important in tumor progression including MB [58]. STAT3 activation may be an important oncogenic factor in MB, and B7-H3 likely has a relationship to STAT3 activity [59,60]. As observed from the exosome mass spectrometry data, STAT3 is found in exosomes isolated from B7-H3 OE D283 cells. Interestingly, without STAT1 being found in these exosomes, STAT3 activity and B7-H3 expression may indicate a more malignant, aggressive MB genotype. Indeed, STAT3 has been characterized as a tumorigenic transcription factor [61,62]. For instance, it induces HIF-1α expression, priming tumor cells for a hypoxic microenvironment [63]. Additionally, work on myeloma cell lines showed that B7-H3 activates STAT3, subsequently promoting cell proliferation, by encouraging the degradation of SOCS3, a known inhibitor of STAT3 phosphorylation [60]. While the functional significance of the observed differential packaging of distinct STAT isoforms between the control and B7-H3 OE exosomes has not been tested, further investigation could lead to valuable insights into the downstream effects of these proteins on the tumor microenvironment.

Our mass spectrometry data also revealed a strong association with enolase isoforms. α-enolase (ENO1) and β-enolase (ENO3) were present in B7-H3 OE exosomes while absent from D283 control exosomes. Additionally, B7-H3 OE exosomes also contained enolase 4. Interestingly, hexokinase 1 and 2 (HK2) were also observed in both control and B7-H3 OE exosomes. A previous publication investigated the relationship between B7-H3 and ENO1, finding that downregulation of B7-H3 leads to a downregulation of ENO1 and subsequent impairment of glycolysis in HeLa cells [64]. Another study indicated that B7-H3 may promote glycolysis by upregulating HK2 [65]. Hexokinases are the initial rate-limiting enzymes in glycolysis while enolases are the penultimate enzyme in glycolysis, strongly implicating B7-H3 in glycolysis [66,67]. Our results support these previous observations but also reveal a possible route by which B7-H3 influences local, and potentially distal, tumor metabolism. Future investigation into the role that B7-H3 may play in glycolysis could focus on discerning how B7-H3 signaling leads to ENO upregulation intracellularly or whether there are protein complexes that form between B7-H3 and ENO isoforms.

As observed in our mass spectrometry analysis, PDGF signaling molecules pathway were highly represented in B7-H3 OE exosomes. Major pathway molecules such as PDGFRA and PDGFRB were observed in both control and OE exosomes. PDGF receptor molecules and B7-H3 have both been found on the surface of human mesenchymal stem cells (MSC) [68]. PDGF signaling has been found to be important in MSC and osteoblast migration during bone development [69]. PDGF signaling, as well as TGF-β pathway activity, can enhance BMP pathway activation, leading to bone cell migration and proliferation [70,71]. Previous studies have shown a relationship between TGF-β and B7-H3 indicating that TGF-β may upregulate B7-H3, leading to classical B7-H3-mediated immune evasion [72]. Perhaps B7-H3 has an unexplored relationship with the BMP signaling cascade that is a mechanism for B7-H3-influenced metastasis. Research into bone cancers and related disorders have explored the role that B7-H3 may have in their progression, predominantly focusing on the relationship between B7-H3 and MMP activity, but our results indicate that B7-H3 could influence bone disorders by enhancing BMP signaling, leading to hyperactive bone cell proliferation or migration [73]. This potential relationship could be a focus for future studies regarding B7-H3, osteosarcomas, and other cancer-related signaling pathways.

As noted previously, B7-H3 may be a useful prognostic marker for MB patients [17,74]. Interestingly, various cancer models have shown that increased microvesicle and exosome shedding correlates with more aggressive neoplasms [75,76,77,78]. Since B7-H3 appears to increase exosomal production in D283 cells, further investigation into this relationship may lead to improved understanding of how valuable B7-H3 may be as a prognostic marker. Additionally, B7-H3 overexpression led to an increase in the size distribution of D283-derived exosomes, suggesting that these exosomes have greater protein carrying capacity. Increased exosome size would then allow for enhanced loading of B7-H3-associated pathogenic molecules such as MMPs and chemokines, as observed from our mass spectrometry results.

## 4. Methods

### 4.1. Antibodies and Reagents

Antibodies for B7-H3, CD-63, p-STAT1, STAT1, and Actin were purchased from Santa Cruz Biotechnology Inc. (Dallas, TX, USA). B7-H3 overexpression plasmid (B7-H3 OE) was purchased from Sino Biological Inc. (Wayne, PA, USA, HG-11188).

### 4.2. Cell Lines and Transfections

The D283 Med (ATCC), D425 Med, D458, UW228, and ONS76 cells (kindly gifted by Dr. Rajeev Vibhakar, University of Colorado) were grown in complete DMEM media (10% FBS, 1% penicillin/streptomycin, 1% Sodium Pyruvate) (Gibco, Waltham, MA, USA). Human endothelial cells, HMEC, were grown in Medium 200 (Gibco, Waltham, MA, USA) supplemented with low serum growth supplement (Gibco, Waltham, MA, USA). All cell lines were incubated at 37 °C with 5% CO_2_. Transfections for D283 cells were carried out using Lipofectamine 2000 (ThermoFisher, Waltham, MA, USA). D283 cells were serum-starved for 1 h in 4 mL of serum-free media on 100 mm plates prior to transfection. Cells were incubated with the reagent-plasmid complex for 8 h, and then 4 mL of complete media was added for overnight incubation. The following morning, the media was replaced with serum-free media and incubated further for 24 h. Exosome isolation was performed afterwards.

### 4.3. Exosome Isolation and Fluorescent Nanoparticle Tracking (NTA)

The exosomes were isolated, purified, and NanoSight Nanoparticle Tracking Analysis (NTA) analyzed by System Biosciences (SBI) (Palo Alto, CA, USA). The exosome size, distribution, and concentration were measured by NTA analysis. Briefly, 2 uL of purified exosomes were labeled using the ExoGlow-NTA fluorescent labeling kit (Cat #EXONTA100A-1, SBI). Labeled exosomes were diluted by a factor up to 1500 (depending on the sample) in a final volume of 300 uL for NTA using NanoSight LM10 fitted with a 488 nm wavelength laser and 500 nm LP filter (Malvern Instruments, Malvern, UK). Both light-scattering and fluorescence modes were employed to detect particle counts/mL and size distribution of particles in solution. Particle counts/mL and size distribution of both modes were overlaid to generate light-scattering and fluorescent counts to determine the overall fraction of exosomes in the solution. The fluorescence mode reflects exosome-specific particle concentration and size distribution data. We also isolated exosomal fraction using total exosomal isolation kit (Invitrogen, Carlsbad, CA, USA) following the manufacturer’s protocol and used it for the coculture immunoblot experiment (Figure 3D).

### 4.4. Mass Spectrometry and Immunoblotting

The purified and NTA analyzed exosomes that we received from the SBI were used for the mass spectrometry and overlaying experiments. The exosomal fraction were analyzed by LC-MS/MS (University of Illinois, Urbana-Champaign) using a standard protocol [79]. HMEC and UW228 cell lines were incubated with 15 µg/well of D283-derived exosomes in 1.4 mL of SFM for 24 and 6 h, respectively, prior to collection for immunoblot. Immunoblots were performed as described previously [80]. Densitometry analysis was conducted using ImageJ. One-way ANOVA statistical analysis was employed to calculate *p*-values.

### 4.5. F-Actin Staining

The FIT-phalloidin-based F-actin red fluorescence (Abcam, Cambridge, UK) staining was done on D283, D425, and D458 cells on 3-well slides (ibidi, Fitchburg, WI, USA) to label cell membrane-associated cytoskeletal elements. The cells were grown for 24 h, fixed and stained according to the vendor’s instructions. Calcein AM (2 µM) green fluorescence (Invitrogen) was used to stain D283 cell-derived B7-H3 OE exosomes. Isolated exosomes were diluted in PBS and coincubated with Calcein AM (0.5 µL of Calcein/15 µg of exosomes) at room temperature for 30 min. The Calcein AM stained exosomes (5 µg) were then incubated for 6 h on to the MB cells containing 1 mL of the complete media. The cells were fixed and stained with F-actin. The images were captured with an Olympus BX61 Fluoview confocal microscope (Olympus, Center Valley, PA, USA) at 40× magnification. The negative control to confirm nonspecific Calcein AM staining was done using SFM (processed in the same way as exosome extraction) overlaid on D458 cells followed by F-actin staining

## 5. Conclusions

Overall, our results indicate a novel role for B7-H3 in MB tumor progression. B7-H3 induces greater exosome secretion and stimulates increased exosome size in D283 MB cells. Further, upregulation of B7-H3 expression intracellularly also increases the presence of B7-H3 in exosomes secreted from these cells. B7-H3 associates with a variety of proteins in D283 exosomes including STAT3, AKT2, MMP2, and MMP9. B7-H3 also has an association with a variety of enolase isoforms and hexokinases, indicating a potential relationship with MB glycolysis. Increased B7-H3 expression in D283 exosomes can also increase B7-H3 expression ectopically in UW228 cells, a MB cell line that differs in subgroup classification. Future studies may focus on determining the specific mechanisms by which B7-H3 may induce exosome production and secretion and whether Class II PI3Ks are involved in this process. Additionally, B7-H3 may influence the tumor microenvironment, in part, from its role in exosomal packaging and production. Further elucidation of the role of B7-H3 in MB exosome production will lead to a greater understanding of MB tumor progression.

## Figures and Tables

**Figure 1 ijms-21-07050-f001:**
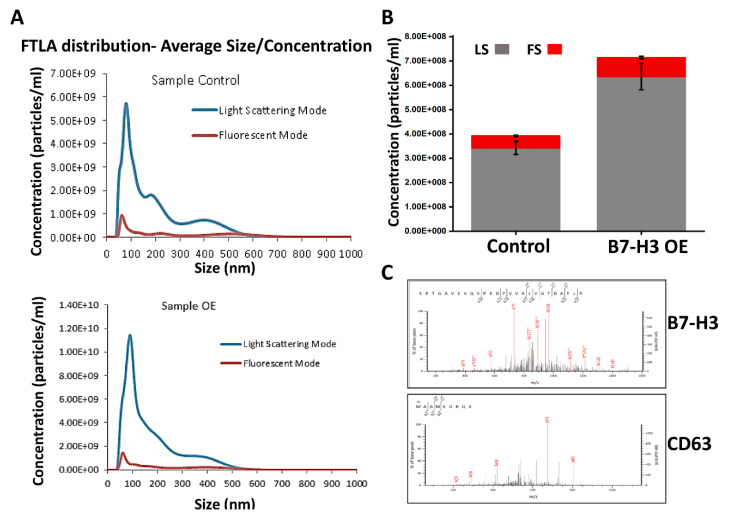
B7-H3 increases exosome secretion in medulloblastoma (MB) cells: (**A**) graphs from fluorescent NTA (SBI) showing increased size distribution of exosomes between control and B7-H3 overexpressing (B7-H3 OE) D283 cells. (**B**) Bar graph showing increased absolute concentration of exosomes in B7-H3 OE conditioned media compared to control cells. Error bars shown for light scattering mode and fluorescent mode. (**C**) LC-MS/MS analysis showing the presence of B7-H3 and CD63 peptide peaks in exosomes isolated from D283 cells.

**Figure 2 ijms-21-07050-f002:**
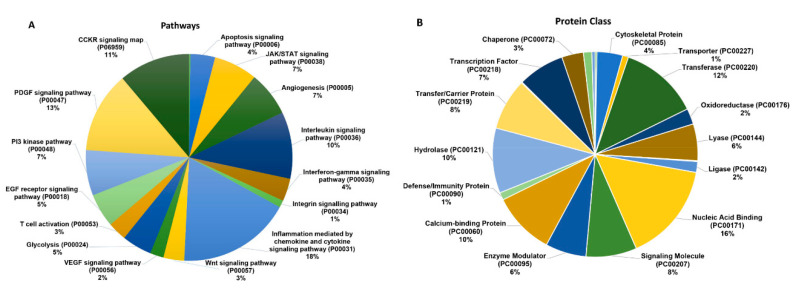
B7-H3 is associated with various cancer signaling pathways in MB exosomes: (**A**) results from mass spectrometry conducted on exosomes derived from B7-H3 OE D283 cells. Pie chart of mass spectrometry data uploaded to PANTHER (pantherdb.org) for annotation. GO annotation for molecules associated with each major cell signaling pathway is visualized. (**B**) Pie chart of mass spectrometry data annotating molecules based on protein class. Percentages indicate percent of all molecules assessed via LC/MS-MS that make up the enriched pathway.

**Figure 3 ijms-21-07050-f003:**
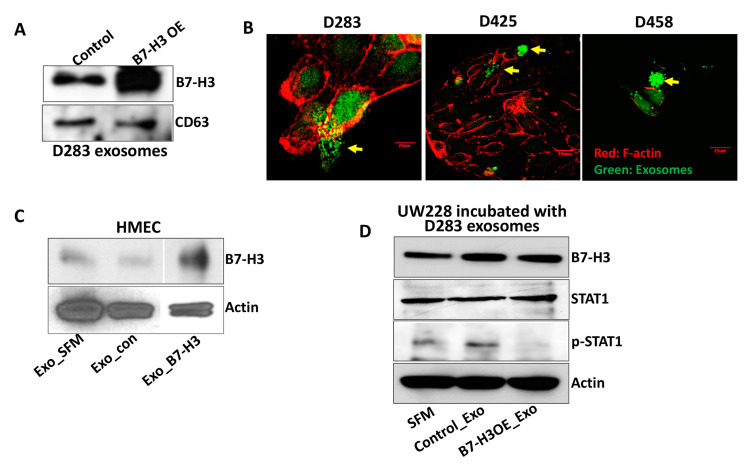
B7-H3 expression can be induced via MB-derived exosomes in recipient cells: (**A**) Western blot of isolated exosomes (5 µg/lane) from D283 cells showing increased B7-H3 presence in B7-H3 OE transfected cell-derived exosomes compared to control. CD63 used as exosomal loading control. (**B**) F-actin staining (red) of D283, D425, and D458 cells incubated with D283-derived B7-H3 OE exosomes (green). Briefly, 15 µg of exosomes (5 µg/well) were incubated with 0.5 µL of Calcein AM in SFM for 30 min in an Eppendorf tube and then overlaid on a 3-well chambered slide (ibidi, Fitchburg, WI) with 1 mL media containing D283, D425, and D458 cells for 6 h. Yellow arrows indicate likely areas of stained exosomes fusing with target cells. (**C**) Western blot of HMECs incubated (24 h) in SFM alone, with D283 control exosomes, or D283 B7-H3 OE exosomes showing increased B7-H3 expression intracellularly. Actin was used as a loading control. The B7-H3 and Actin blots shown here are cropped using PowerPoint for clear representation. Lane 3 from the original blots of both B7-H3 and Actin were cropped from the figure as it is not relevant to this study. Lane 4, which represents exosomes from B7-H3 overexpression (EXO_B7-H3 OE), is now represented as a single lane with actin as a loading control. The corresponding uncropped full-length blot is included in Appendix A. (**D**) Western blot of UW228 cells incubated (6 h) with SFM alone, D283 control exosomes, or D283 B7-H3 OE exosomes showing increased B7-H3 expression along with decreased STAT1 phosphorylation. Actin was used as a loading control.

**Table 1 ijms-21-07050-t001:** Exosomal Mass Spectrometry Molecules.

Control EV			B7-H3 OE EV		
Gene	Associated Pathway(s)	Matched Peptide Sequence	Gene	Associated Pathway(s)	Matched Peptide Sequence
STAT1	JAK/STAT, IFN-γ/IL-12, Chemokine signaling	DPIQMSMIIYSCLKE	STAT3	JAK/STAT, Chemokine signaling, Stem cell/LIF, IL-6/HIF-1	NQGVPVLIVANK
STAT2	JAK/STAT, Chemokine signaling	LSLDLEPLLKAGLDLGPELE	c-MYC	MAPK, PI3K/AKT, WNT, TGF-β	KQIVAGVNYFLDVE
MYCN	Group 4 MB	DAPPQKKIK	AKT2	MAPK, HIF-1, PI3K/AKT, Chemokine signaling, VEGF, Ras	VSLAKPKHRVTMNE
IKKB	NF-κB, MAPK, mTOR, PI3K/AKT	AAMMNLLRNNSCLSKMK	MMP2	Endothelial migration/angiogenesis, MAPK/ERK, Myc	NVAADIAVQLCE, VWELGGCANKE
TANK	NF-κB, NOD-like receptor	GPQQPIWKPFPNQDSDSVVLSGTDSE	MMP9	TNF, IL-17/MAPK, NF- κB, angiogenesis	NKPTRPVIVSPANETME
TGFB1	TGF-β	LLAPSDSPEWLSFDVTGVVR, RGDLATIHGMNRPFLLLMATPLER	TIMP2	MMP2/MMP9	FTTSVVRR
TGFR1	TGF-β	VLDDSINMK	NFKB2	NF-κB, MAPK, PI3K/AKT, Ras	LAPASPMASPGGSIDERPLSSSPLVRVK, LLTDVQLMK, VVNKLIQFLISLVQSNR
CCL2	JAK/STAT, Src, MAPK, PI3K/AKT, NF-κB	ICADPKQKWVQDSMDHLDK	IL2	PI3K/AKT, JAK/STAT, MHC/Antigen signaling, ZAP70	HPRNIQESPF
H2A1D	Histone 2 complex	VGAGAPVYLAAVLE	CCL5	JAK/STAT, PI3K/AKT, TNF, TLR4, LPS/ERK	SSTLIGR
			CCR9	CCL25, PI3K/AKT, JAK/STAT, RhoA/ROCK, MAPK	LEVLQDCTFE
			TSG101	ESCRT complex, vesicle formation	AMLASRSASLLK
			H2A1B	Histone 2 complex	VGAGAPVYLAAVLE

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
