# Peer review of "B7-H3 in Medulloblastoma-Derived Exosomes; A Novel Tumorigenic Role"

_ijms, 2020, doi:10.3390/ijms21197050_

Round 1

Reviewer 1 Report

The manuscript describes a specific molecular characterization of medulloblastoma-derived exosomes.

The topic is quite interesting with potential downstream applications.

Some technical improvment of experimental data are however required.

In general, all immunoblotting data need densitometric evaluation to support statements as " .... We observed a significant (p≥0.001) increase in the levels of B7-H3 in the B7-H388 OE cells compared to the control (Supplementary Fig 1A)...."

In addition, the quality of some immunoblotting evidences are quite poor: for example, actin bands are sometimes in saturation levels (supp fig 1A, HMET) or sometimes not convincing me (the quite perfect sharp band on Fig 3C Exo B7_H3). Furthermore, p-STAT1 bands are of very low quality, as documented in Figure Suppl 3 D (I wonder how to define the expected 91 kD).

For experiment using green labelled exosome, in the Methods, there`s no indication on a purification of unspecific fluorescent dyes (it`s necessary to adopt column purification kits). In relation to this potential contamination of fluorescent dyes, Fig 3B does not include a negative control of cells + column-purified dyes. In this Figure, no micrometric scale bars are reported.

In conclusion, Figure 1C and 2 labels have to be improved in quality.

Overall, this manuscript requires major revisions to be further considered.

Author Response

The topic is quite interesting with potential downstream applications.

Some technical improvement of experimental data are however required.

In general, all immunoblotting data need densitometric evaluation to support statements as " .... We observed a significant (p≥0.001) increase in the levels of B7-H3 in the B7-H388 OE cells compared to the control (Supplementary Fig 1A)...."

We have added densitometry evaluation of the western blots as recommended to show the significance of proteins expressed (see lines 184, 198, 201).

In addition, the quality of some immunoblotting evidences are quite poor: for example, actin bands are sometimes in saturation levels (supp fig 1A, HMET) or sometimes not convincing me (the quite perfect sharp band on Fig 3C Exo B7_H3). Furthermore, p-STAT1 bands are of very low quality, as documented in Figure Suppl 3 D (I wonder how to define the expected 91 kD).

As per the reviewer’s suggestions, we performed western blot analysis and have completely replaced Figure 3A, 3D and supplementary Fig. 1A. Figure 3C blots were cropped and replaced in the revised manuscript. The original blots for Figure 3C are kept in Supplementary Figure 3.

For experiment using green labelled exosome, in the Methods, there`s no indication on a purification of unspecific fluorescent dyes (it`s necessary to adopt column purification kits). In relation to this potential contamination of fluorescent dyes, Fig 3B does not include a negative control of cells + column-purified dyes. In this Figure, no micrometric scale bars are reported.

The methods section has been changed to reflect more details of the Calcein staining experiment (see lines 363-368). We agree that a negative control is necessary to rule out non-specific staining of cells with the Calcein reagent. For this we stained SFM media (processed the same way as exosome extraction) with Calcein AM (please see methods section) and overlayed on D458 cells. The negative control for non-specific Calcein AM staining was performed and the images are represented in Supplementary Figure 2.

In conclusion, Figure 1C and 2 labels have to be improved in quality.

We have re-labeled the figures and have rewritten the recommended portions of the figure legends (see lines 106 and 124) to further clarify the data in each figure.

As the Reviewers and Editor suggested, we made all the relevant corrections in this revised manuscript. We sincerely feel that the substantial changes that we have made to address these points have significantly strengthened the paper. We request that the Editor kindly consider this manuscript for Publication.

I hereby confirm that the content of this manuscript is original and that it has not been published or accepted for publication, either in whole or in part, in any form. 

Reviewer 2 Report

Authors reported well written and excellent organized research paper. The aim of the study was to investigate the role of B7-H3 in MB progression.

Major concerns;

  1. Please, report ckear purpose of the study at the end of the section "Introduction"
  2. Please, describe in detail the blood sampling and a way of the sample before labeling of exosomes.

Minor concerns:

  1. Please, structurize the abstract giving the aim, methods, resuls, and conclusion separately.
  2. Please, add one or two sentences in the section "Inroduction" to clear explane the nomenclature of the extracellular vesicles and the role of the exosomes.
  3. Fig 2 requires to be modified. Fig 2A and Fig 2b should be replaced in vertical position to be easily recognized.
  4. The sentence "B7-H3 present in cancer cell exosomes may play an important role in cell-cell communication and tumor cell signaling in addition to its role in immuno-evasion." (line 146-147) is a hypothesis of the study and should be replace at the section "Introduction"
  5. The conclusive part may contain the sentence from the section 2.3 "B7-H3 may play an active role in exosome biogenesis and exosome
    mediated pathogenicity in MB." instead of "our results indicate a novel role for B7-H3 in MB tumor progression"
  6. Section "Study limitations" is required

Round 2

Reviewer 1 Report

Dear Authors

Manuscript ID ijms-918152 B7-H3 in medulloblastoma-derived exosomes; a novel tumorigenic role    

after a first round of reviewing, I realize all my suggestions/criticism have been favourably considered in the revised form of the manuscript. I therefore suggets the Editor to accept the emended manuscript for publication.